# Automated Low-Cost Soil Moisture Sensors: Trade-Off between Cost and Accuracy

**DOI:** 10.3390/s23052451

**Published:** 2023-02-22

**Authors:** Dimaghi Schwamback, Magnus Persson, Ronny Berndtsson, Luis Eduardo Bertotto, Alex Naoki Asato Kobayashi, Edson Cezar Wendland

**Affiliations:** 1Division of Water Resources Engineering, Department of Building and Environmental Technology, Lund University, Box 118, SE-221 00 Lund, Sweden; 2Department of Hydraulics and Sanitation, São Carlos School of Engineering, University of São Paulo, CxP. 359, São Carlos 13566-590, Brazil

**Keywords:** low cost, water content, soil moisture, data collection, capacitive sensor, smart agriculture

## Abstract

Automated soil moisture systems are commonly used in precision agriculture. Using low-cost sensors, the spatial extension can be maximized, but the accuracy might be reduced. In this paper, we address the trade-off between cost and accuracy comparing low-cost and commercial soil moisture sensors. The analysis is based on the capacitive sensor SKU:SEN0193 tested under lab and field conditions. In addition to individual calibration, two simplified calibration techniques are proposed: universal calibration, based on all 63 sensors, and a single-point calibration using the sensor response in dry soil. During the second stage of testing, the sensors were coupled to a low-cost monitoring station and installed in the field. The sensors were capable of measuring daily and seasonal oscillations in soil moisture resulting from solar radiation and precipitation. The low-cost sensor performance was compared to commercial sensors based on five variables: (1) cost, (2) accuracy, (3) qualified labor demand, (4) sample volume, and (5) life expectancy. Commercial sensors provide single-point information with high reliability but at a high acquisition cost, while low-cost sensors can be acquired in larger numbers at a lower cost, allowing for more detailed spatial and temporal observations, but with medium accuracy. The use of SKU sensors is then indicated for short-term and limited-budget projects in which high accuracy of the collected data is not required.

## 1. Introduction

Soil water content (SWC), amount (volume or mass) of water per dry soil, is a key variable that governs important processes such as evapotranspiration, runoff, and groundwater recharge [1]. In this context, the soil water content (%) is a key element in, e.g., precision agriculture, deficit irrigation, and control of soil physicochemical processes [2,3]. The soil heterogeneity affects the spatial behavior of the vertical (re)distribution of water in the vadose zone [4]. Water movement and storage are strongly influenced by land use and cover (leaf characteristics, water demand, depth and typology of roots), pedology (porosity, density, granulometry, and infiltration capacity), and climatologic properties (rainfall depth, duration, and intensity, minimum and maximum air temperature, solar radiation, sunlight duration and wind direction and speed) [5]. Hence, soil moisture monitoring is essential to achieve a better understanding of processes that are dependent on the soil–vegetation–atmosphere interaction [6], all of which are needed for sustainable soil and water resources management.

Soil water content can be measured by direct or indirect methods. Direct methods consist of separating the water from the soil with solutes and chemical reactions, or, more traditionally, through oven-drying. Although the latter is the most accurate method to measure soil moisture, it is destructive and does not allow real-time measurements [7]. On the other hand, a wide range of non-destructive indirect methods exist, allowing soil moisture monitoring with high accuracy, ranging from remote sensing [8] to the point scale [9]. Additionally, soil water content monitoring for field-scale applications, such as irrigation, demands non-destructive real-time monitoring with extensive spatial coverage that can be costly.

In this context, several local-scale soil moisture sensors, commercial and in development, with different operation principles are available. Among them, we highlight the neutron probe [10], which operates with the neutron thermalization principle; time domain reflectometry (TDR) [11], time domain transmission (TDT) [12], and capacitive sensors [13] that use soil electromagnetic properties to quantify the water content, heat pulse probes [14] based upon the soil heat transfer properties, and optical sensors [15] that use near-infrared reflectance techniques. A detailed description of the state of art of methodologies used for soil moisture monitoring is given by [16,17,18]. Commercial sensors provide soil water content observations with very high accuracy, but due to the high acquisition and maintenance costs (thousands of dollars), elaborate and time-consuming procedures during calibration, and highly qualified personnel, extensive applications in field monitoring networks are limited. Alternatively, there are several types of low-cost sensors, such as the YL-69 [19], STEMMA [20], and the SKU:SEN0193 [6,21], that provide soil water content data with acceptable accuracy for projects under limited financial resources. However, the question remains, how should we balance low cost with high, or reasonable, accuracy when selecting the most suitable sensor to be employed in a project?

The main objective of this study was to evaluate the use of low-cost technologies with open-source characteristics to monitor the wetting front in soil under laboratory (calibration and infiltration column) and field conditions. We focused on the capacitive sensor identified as SKU:SEN0193, a low-cost soil moisture sensor that operates with low power consumption, being ideal for isolated conditions [22] while compatible with common types of microcontrollers (Arduino and Raspberry). Due to these characteristics, the sensor has gained recent attention from the scientific community [3,6,22,23,24,25]. We contextualize the trade-off between cost and accuracy in a qualitative comparison to common commercial sensors.

## 2. Materials and Methods

### 2.1. SKU Sensor Description and Theory

We used the SKU:SEN0193 capacitive sensor (Figure 1a, hereafter called SKU), controlled by an Arduino Uno. The SKU consists of two coplanar conductive plates directly isolated, separated, and surrounded by the soil material. The soil is a multiphase material composed of solid (mineral particles), liquid (water), and gas phases [6]. Some advantages of this sensor are that it is not affected by the presence of salts commonly used during the fertilization of crops [3], and it is Arduino compatible, a free software and hardware low-cost microcontroller. Additionally, the sensor operates under low voltage (3.3 to 5.5 V), a particularly important characteristic for field applications. The device output data are expressed through frequency oscillation, commonly between 260 Hz (high soil water content) and 520 Hz (low soil water content), once the application of an electric voltage creates an electric field that varies with the change in soil water content. A more detailed description of the SKU sensor’s function is given by [6].

The low acquisition cost of SKU sensors comes with several disadvantages, e.g., they are fragile as their electronic components are exposed to light, heat, and air humidity. This might limit their use to simple tests in laboratory benches and plant pots. The SKU has a required optimal soil contact area (Figure 1a), and since we intended to use it under different environmental conditions, it was necessary to protect the upper part of the sensor to avoid physical damage, short circuit, and oxidation. Thus, this portion of the sensor was waterproofed with enamel, wrapped with heat shrink, protected with a plastic case, and later filled with silicone (Figure 1b).

### 2.2. Initial Tests and Calibration

Several initial tests were carried out to investigate the sensor’s response under different input voltage, temperature, and soil moisture conditions. Laboratory tests were conducted with disturbed soil samples collected in an area where the field tests were later carried out. The soil is a Quartzipsamments, with a sandy texture, good drainage, acidic, and poor in nutrients [26]. Table 1 presents the texture, cation exchange capacity, organic matter, bulk, and particle density for different depths of the study area. Although we recognize that for more accurate results, the calibration process must be performed with soil samples from the depth at which the sensor will be in contact [6,22], we only used samples from the 30 cm depth for laboratory tests. The calibration process (Figure 2a) comprised tests of more than 63 sensors, and thus the use of different soil samples would interfere with the comparison of the sensor’s outputs. Furthermore, at our field site, the soil’s physical parameters are fairly constant with depth.

The calibration was performed using 11 soil samples, each with a volume of 600 cm^3^ with known volumetric soil water content (SWC, volume of water divided by the dry bulk soil volume, expressed in m^3^/m^3^), varying isometrically between dry and saturated conditions. Soil samples were dried at 105 °C for 48 h (dry condition), mixed with water to reach the target SWC and compacted until reaching the bulk density found in the field. Saturated condition means that all the pores (empty spaces between the solid soil particles) are completely filled with water.

To assess the influence of temperature on the sensors’ output signal, we performed a laboratory test under well-defined thermal conditions (Figure 2b). We prepared three 600 g soil samples, all with a SWC of 0.18 m^3^/m^3^, a value commonly found in the field. Three soil samples were prepared, each equipped with an SKU sensor and a thermocouple. The thermocouple was positioned near the middle of the SKU sensor. The thermocouples monitored the temperature with a resolution of 0.1 °C and a sampling interval of 60 s during the test. The soil moisture sensors were connected to an Arduino Uno shield, a datalogger shield, and a Relay shield, all powered by a power bank unit. Each sensor was programmed to perform five consecutive measurements of soil moisture every 5 min. The experimental setup was exposed to the following consecutive conditions; kept during two hours at room temperature (20 °C); one hour inside a refrigerator (2 °C); one hour at room temperature; one hour inside an oven with a controlled temperature (32 °C); and four hours at room temperature. During the whole test, an extra thermocouple exposed to the air was kept aside the soil sample. Although the test did not last as long as the diurnal temperature cycle, we aimed to create field-scale thermal conditions to evaluate the sensors’ response to a similar temperature variation. Furthermore, a longer test could lead to undesired sample evaporation. 

### 2.3. Laboratory Test: Infiltration Column

The second phase of the sensor testing aimed to analyze its ability to identify the wetting front arrival during an infiltration or rainfall event (Figure 3). The test was carried out in a 125 mm diameter PVC column filled with dry soil, creating a homogenous condition. Water was applied at a constant rate of 9.88 mm/h until the water reached the bottom of the column. Ten sensors were inserted into the soil column spaced at 10 cm. Their output was collected in five replicates at an interval of two minutes using three Arduino microcontrollers. Time acquisition and data storage in a. txt file type were enabled through a datalogger shield attached on the top of the Arduinos. As a power supply system, we used a 5–12 V output source. The list of materials and their costs are given in Table 2.

### 2.4. Wetting Front Simulation

To investigate the sensors’ ability to represent the wetting front arrival in the soil column, we applied the Hydrus-1D package [27] for the theoretical representation of water vertical flow. Hydrus is a computational package developed by the Salinity Laboratory of the US Department of Agriculture to simulate the transport of water, solutes, and heat in one, two, or three dimensions in a saturated or unsaturated porous medium. The software uses the finite element method to numerically solve the Richards Equation [28] and describe saturated/unsaturated flow. Additionally, it uses the van Genuchten equation [29] to describe the soil water retention curve, which relates the potential pressure to the hydraulic conductivity [30]. For describing the hydraulic conductivity, it uses the equation of Mualem [31]. The van Genuchten hydraulic parameters ($\theta_{r},\;\theta_{s},\;\alpha ,\;n,$ and $K_{s}$) were estimated using the neural network prediction module Rosetta [32] using the granulometric material percentage and bulk density. The upper conditions of the soil profile correspond to atmospheric boundary conditions (BC) under constant incoming water flow, and a lower free drainage. Table 2 contains the input values and conditions used for the parameters while simulating the infiltration process in Hydrus-1D.

### 2.5. Field Tests and Study Area Description

The sensors were tested under field conditions. The tests took place in an experimental 100 m^2^ field plot (5 m wide by 20 m long) without vegetation cover and delimited with metal sheets approximately 30 cm high (see Figure 4). The plots were installed in 2011 [33,34,35] and surface runoff and erosion have been monitored continuously [36,37]. The experiment was carried out at Instituto Arruda Botelho (IAB), Itirapina, central region of the State of São Paulo, Brazil (latitude 22°10′ S, longitude 47°52′ W, elevation of 790 m). The region has an average annual rainfall of approximately 1,500 mm, with a rainy season between October and March [38] and, according to the Köppen–Geiger classification system, the climate is humid subtropical (Cwa), with hot and rainy summers and cold and dry winters [39].

To collect and store data, a low-cost monitoring station was designed using Arduino microcontroller, capacitive soil moisture sensors, datalogger shield, relay shield, solar panel, charge controller, step-down, and battery (Figure 5). The output data from the sensors installed in April, 2021 at depths of 10 (SR1), 30 (SR2), 60 (SR3), and 90 cm (SR4) were collected in five replicates at an interval of 2 min by the Arduino microcontroller. Time acquisition and data storage were enabled through the datalogger shield while the relay shield allowed current to pass to the sensors only at the time of data collection, saving power and expanding the sensors’ lifetime. As a power system, the controller uses a reduced energy voltage (7 V) through a step-down component from a 12 V/7 A battery supplied daily by a 60 W solar panel. The code used for the microcontroller is publicly available at [40]. Table 3 presents the list of equipment used in the construction of a low-cost monitoring station composed of a self-powered system for measuring soil moisture at four soil depths. The station had a total cost of BRL 870 (USD 163), a significantly lower budget compared to other standard technologies, such as FDR and TDR sensors, which would have cost up to BRL 16,000 (USD 2990). Additionally, capacitive sensors and other electronic components can be easily found at specialized electronic stores.

## 3. Results

### 3.1. Laboratory Tests: Soil Moisture, Temperature, and Voltage

The SKU sensor is part of a set of a generation of open-source hardware developed for operation through low-cost microcontrollers and thus has two power options: 3.3 V and 5.5 V. Figure 6 shows sensor output from one SKU sensor versus soil water content for both input voltages. The sensor output frequencies are highly dependent on the input voltage with higher output for higher voltage. The best fit between sensor output and soil water content was found for the 3.3 V option (R^2^ = 0.871) compared to the 5.5 V option (R^2^ = 0.798). While performing these initial tests, we noted different sensor response speeds: when immersed under water, sensors can respond immediately (step type), or gradually (slope type).

After identifying the power supply (3.3 V) that produced the best correlation between SWC and sensor output, we started the tests to construct the calibration curves. A higher accuracy can be achieved using the individual calibration of each sensor (all sensors with R^2^ above 0.94). As identified for other sensors in Figure 7, some sensors had difficulties in measuring soil water content above 0.3 m^3^/m^3^, while SR1 had similar difficulty below 0.1 m^3^/m^3^. A possible explanation for this is that the soil samples were not completely compacted for dry and near-saturation conditions.

Individual sensor calibration is very time-consuming. To save labor, we tested a simplified calibration based on the average slope of the sensor output-SWC relationship plus sensor output when inserted into dry soil: equal to or below 460 Hz (group 1) or above 460 Hz (group 2), as given in Figure 8. To simplify calibration and provide a better correlation between sensor readings and actual SWC, we developed a universal calibration curve (Figure 9) based on data from 63 sensors. Table 4 comprises statistical metrics comparing the observed and predicted SWC for each calibration method (individual sensor, single-point, and universal). Despite obtaining a satisfactory linear correlation (R^2^ = 0.949) between known SWC and the sensor output frequency, the response amplitude was very high, especially at the end of the curves (dry and saturated soil), as also found by [21]. Under saturated conditions, 50% of sensor outputs were within the range between 215 and 295 Hz, wider than the range observed under medium SWC level (0.12 m^3^/m^3^), 325 to 370 Hz. Thus, due to the high variability identified for universal and single-point calibration curves, the construction of individual calibration curves is suggested (Figure 7). 

Figure 10 shows the performance of the sensor during the temperature dependency test. The air temperature measured by the thermocouple ranged from 2.7 °C to 32.1 °C while the soil temperature was between 11.8 and 32.3 °C. These values were similar to the range found in the field tests, between 4.2 and 34.3 °C. When the sample cools with the decrease in air temperature, there is an increase in the sensor frequency output. The same pattern of inverse correlation between sensor output and soil temperature occurs when heating the sample. For a temperature change of 20 °C, there was a change in the soil water content of approximately 0.015 m^3^/m^3^ meaning 4% of soil saturation spectrum. So even though the temperature dependency is clear, the effects are small.

### 3.2. Soil Column Tests

Figure 11a shows the infiltration curves over time for the different depths. The infiltration rate varied among the layers and it was higher in the top layer (10.7 cm/h) than in the bottom (6.1 cm/h), possibly associated with lower soil compaction near the surface. It took approximately 14 h from the test start to the last sensor, located 100 cm from the top of the soil column, to detect the wetting front arrival. It is important to highlight the sharpness of the infiltration curves identified by the sensors, indicating the exact moment of wetting front arrival and not so much about how long it took to completely pass. For comparison of monitored data, the wetting front was estimated through Hydrus (Figure 11b) based on pedological data and incoming water flow (Table 2). No compaction level differentiation was added to the model, and thus the layers presented the same infiltration velocity (7.24 cm/h), which compromises individual comparisons between the measured and simulated wetting front arrival over time. However, when we compared the measured and simulated average infiltration velocities, they were close: 7.14 and 7.24 cm/h, respectively. The infiltration rate was estimated through the sharp oscillations in the sensor output and computed as the ratio between the distance of sensors and time taken between the curves. The Hydrus calibration process based on the observed data comprised many tests using different combinations of target pedological variables and after trials, the best correlation with the observed data was obtained through the calibration of the variables α, n, and $K_{s}$. When we look at Figure 11c presenting the SWC curves after the calibration process using these variables, the main consideration concerns the drastic increase in infiltration speed (8.13 cm/h), which was now is higher than the observed mean.

### 3.3. Field Tests

The last phase of sensor testing aimed to evaluate their applicability under field conditions (Figure 12). The monitored period comprised precipitation events of in total 33 mm occurring between 26 May and 24 July 2021. We performed the analysis at two temporal resolutions: seasonal and daily.

The seasonal oscillations show SWC varying between 0.22 m^3^/m^3^ and 0.31 m^3^/m^3^ (82% soil saturation) near the surface (10 cm depth–SR1) while smaller oscillations were observed at the other depths (0.24 m^3^/m^3^ to 0.3 m^3^/m^3^). SR1 responded almost immediately to the occurrence of precipitation events (e.g., 22 May and 10 June), indicating soil drying during the following days. The precipitation event that occurred on 22 May had a high intensity and low duration (7.8 mm lasting 1 h and 30 min) that, added to the non-vegetation condition of the study area, favored surface runoff. Thus, sensors located at other depths did not identify the presence of a wetting front passage as shown by SR1. On the other hand, the June 10th event was intensive with long duration (28.7 mm over 9 h and 40 min) to promote infiltration that was monitored by the sensors at all depths.

Additional to the seasonal oscillations, we observed daily oscillations occurring at SR1 and in smaller magnitudes at SR2, SR3, and SR4. Due to its proximity to the surface, the water content in the soil is evaporated by solar radiation during the day, while during the night, the lower layers supply moisture to the upper one, restoring the previously measured SWC, creating up- and downward cycles. The oscillations identified can be related to the variation of the air temperature (as described in Figure 10), but with little significance since the depths of the sensors are inversely proportional to the daily oscillations, as also observed by [21].

## 4. Discussion

### 4.1. Sensor Performance

Setting up soil moisture observation systems in research projects often means the use of commercial equipment with high acquisition costs. In this study, we present the applicability of a low-cost sensor (Capacitive Sensor SKU:SEN0193 v1.2) for SWC monitoring for lab and field conditions, being able to identify wetting front arrival and daily fluctuations in SWC at a sufficient accuracy for most applications. [41] list 23 unsolved problems for the direction of research related to water resources. The present study relates to question #16 in describing innovative technologies to measure surface and subsurface properties, states, and fluxes at a range of spatial and temporal scales.

Before field implementation, we conducted several laboratory tests: response speed, supply voltage, temperature dependency, calibration curve, and column infiltration. The results indicated two groups of sensors with fast (step-type) and slow (slope-type) response speeds. For the SKU sensor, we suggest initial tests to be carried out, for the priority of step-type sensors. The slope-type sensors can also be used if data collection occurs after the sensor signal stabilizes. It is worth noting that the calibration curve had a better fit when a single sensor was powered with 3.3 V input voltage (Figure 6). However, [42] found less variability of sensor readings when operating under 5.0 V when testing and calibrating the SKU sensor under different soil types in Brazil. Additionally, low-voltage operation represents a 40% reduction in energy demand and an increase in power system autonomy when monitoring remote areas.

Three types of calibration approaches were evaluated. The best results were obtained using individual calibration. Our findings are specific to the pedological characteristics of the soil sample tested. Previous studies [21,24,25,42] also indicated the importance of constructing local calibration curves since the sensor is sensitive to soil texture and bulk density. [42] for example, tested the SKU sensor under three soil textures and found distinct calibration curves. The individual calibration curves (Figure 7) showed satisfactory consistency between SWC and sensor output with R^2^ greater than 0.94, similar to the correlation found in other studies [2,23,24,42,43,44,45,46] On the other hand, the use of universal and single-point calibration curves presented a high correlation between sensor output and SWC, being viable options for quick interpretation of collected data, since individual calibration is a time-consuming task and demands qualified labor [24]. For example, the individual calibration of a sensor takes approximately 3 h while the single-point calibration only takes approximately 15 min. Nonetheless, we are not sure about how significant the impact that soil texture has on sensor output when applying the universal or single-point calibration curve and future studies are then required.

The column infiltration tests (Figure 11) showed the applicability of the sensors in the identification of wetting front arrival. Additionally, the results from these tests illustrate that the sensors register an abrupt change when a change in the monitoring zone occurs, a behavior that has not been reported in previous research. However, [6] mentioned that because the SKU sensor is designed for operating with low energy consumption, the magnetic field created around the sensor is limited to just a few millimeters. With the arrival of the wetting front in this limited portion of the soil, the sensor records an abrupt change in its electromagnetic field. This explains the shape of the monitored curves and those estimated by Hydrus. From the inverse module of Hydrus, we tried to estimate the pedological variables that aim at the best representation between observed data and that estimated by the one-direction simulation of Hydrus. We identified that due to the unique shape of the observed infiltration curves, the Hydrus does not respond coherently while trying to force a passage of the wetting front in a piston format.

Beyond the calibration and bench tests, other recently published studies have focused on the implementation of sensors in the field and the automation of data transmission. Ref. [21] proposed the use of nonlinear machine-learning techniques (Multiple Linear Regression, K-Nearest Neighbor, Support Vector Regression, and Random Forest) over classical linear regression techniques to calibrate the sensors. The authors implemented automatic transmission of field data collection by adding an Ethernet shield to the raspberry microcontroller. Additionally, the authors of [25] indicated that the device is suitable for measuring soil moisture for agricultural purposes after finding an average error of less than 2% in soil moisture readings based on the results of a two-week field test.

During the field tests (Figure 10), we noticed that the output signal of the sensors was influenced by thermal oscillations. However, there was no observable influence of soil temperature on the behavior of the sensors. This may have occurred due to the difference in the design of hardware components used in the construction of the low-cost capacitive soil moisture sensors. Nonetheless, the soil temperature influence on the performance of the sensor (Figure 10) was not included as a correction factor for the data collected in the field (Figure 12) as it represented an oscillation of 0.64 Hz or 0.001 m^3^/m^3^ in SWC per degree Celsius. This is insignificant compared to other factors, such as the degree of soil compaction used during calibration, which induces a significant difference in the frequency response of the sensor (as also found by [6] and [25]). Daily oscillations in the soil saturation level measured in the field (Figure 12) were observed not only in the top soil but also in the lower layers, with smaller amplitude, a similar pattern identified by [47] using the SoilVUE 10 probe (Campbell Scientific Inc., Logan, UT, USA). In addition to the vertical water movement resulting from solar heating, the authors point out the possibility that the observed oscillations are intensified by overheating the monitoring station and the cables that transmit the signal.

### 4.2. Low-Cost Technologies Applied to Water Resources Monitoring: Possibilities and Challenges

The proper management of natural resources and increase in agricultural potential will only be possible from more reliable information regarding soil, water, and vegetation properties [48]. Despite representing a small fraction of globally available freshwater, moisture influences water storage in the hydrological cycle and is of fundamental importance for hydrological, biological, and biogeochemical processes. By monitoring soil moisture under different vegetation cover, a better understanding of the movement of water through soil layers is achieved [49].

In situ data collection and monitoring of hydrological variables over long periods are scarce, especially in developing countries [50]. The water movement in the vadose zone is poorly monitored and among the different factors associated, due to the high acquisition cost of monitoring equipment that compromises the limited budget available in research. The use of low-cost technologies might overcome this limitation, being accessible to most research centers with budgetary limitations. The sensor described here is not intended to represent comparable accuracy when matched to commercial sensors, but instead to promote monitoring from a new perspective without compromising the appropriate level of accuracy. 

Commercial sensors provide single-point information with high reliability at a high acquisition cost, while low-cost sensors can be acquired in greater numbers with a low investment, allowing for a greater access of spatial and temporal variability of soil moisture, but it normally comes with the tradeoff of low accuracy and/or high labor demand. Table 5 provides a comparison of different commercial sensors based on five variables obtained from the operation manuals: financial acquisition cost (up to USD300); data collection accuracy (up to ±0.08 cm^3^cm^−3^, see Table 6); qualified labor demanded for installation, individual calibration and data collection (based on operational manuals); volume sampled during data collection (up to 7800 mL), and life expectancy (up to 15 years). Even though we do not provide direct comparison with commercial sensors, Table 6 summarizes previous studies addressing the accuracy of soil moisture sensor and as found here and by [24] and [42], SUE sensor has a accuracy between 3 and 8 times lower than commercial ones. The tables do not aim to give a complete picture of all available systems, but only a few typical examples. Since the actual accuracy, financial costs, etc., depend on many variables, we used a qualitative comparison method, grading each system and category 1−5. As expected, the use of components with greater durability and accuracy contributes to the increase in the cost of acquiring sensors. When compared to other sensors, the SKU sensor provides a medium accuracy data collection under a significant low acquisition cost. Nonetheless, it has a highly qualified labor demand as it requires station assembly and physical protection, low sampling volume indicating greater fragility to local interference, and short expectation of life. The use of SKU sensors is then ideal for short-term and limited-budget projects in which high accuracy of the data collected is not required.

In addition to the degree of soil compaction previously reported, our findings face several sources of uncertainty such as the use of only one soil type during the calibration process of sensors and assuming that different depths have the same pedological characteristics, the influence of organic matter content, noise, and loss of analog signal during the data transmission, and low-frequency operation highly sensitive to external interference. Despite the limitations identified, the results given prove that the sensors can identify the temporal relative difference of SWC resulting from natural events, such as solar radiation and precipitation, allowing the visualization of daily fluctuations in humidity and the identification of the wetting front.

As further work, the authors suggest the validation of monitoring carried out in the field to properly verify the sensor fluctuations and accuracy. The validation would also help to estimate the sensors’ physical behavior, expiration date, and reading stability over time under unfavorable weather, as they seem to have fragile hardware components. We also suggest adapting working with different low-cost sensors that create a higher electromagnetic field around itself, avoiding abrupt response due to wetting front arrival and SWC oscillation.

Finally, we observe that it is still necessary to reconcile the cost of construction of the monitoring equipment with its robustness to withstand adverse environmental conditions. The constant need for specialized personnel to repair and replace monitoring station hardware and sensors limits the use of low-cost components. For this, greater efforts should be made to support research aimed at developing and testing of sensors applied to hydrology, since they support the understanding of monitored environmental conditions and the reduction in uncertainties in hydrological models of unmonitored areas.

## 5. Conclusions

In this paper, we addressed the trade-offs between accuracy and acquisition cost between low-cost and commercial soil moisture sensors through the assessment of the capacitive sensor SKU:SEN0193 under lab and field conditions. Commercial sensors promote soil moisture measurements with high accuracy at a high acquisition cost. On the other hand, low-cost sensors, such as the SKU:SEN0193, provide data with medium accuracy at a very low acquisition cost, enabling spatial monitoring through multiple-point measurements. Thus, the use of the SKU:SEN0193 sensor is suggested in projects with budget limitations with short duration where there is a medium requirement accuracy or when the spatial variability of soil water content is considerable.

Laboratory tests indicated that the sensor is temperature and voltage sensitive. The use of 5.5 V as supply voltage for the sensors drastically reduced the correlation between output and SWC, thus we suggest the use of 3.3 V. Soil temperature had a negligible impact on the sensor output: 0.001 m^3^.m^−3^ in SWC per degree Celsius. For field implementation, a low-cost monitoring station was built using Arduino as a microcontroller. The sensors could represent daily and seasonal oscillation in soil moisture resulting from solar heating and precipitation.

Despite the physical fragility of the hardware used (sensors and monitoring station) and the lower accuracy when compared to other commercial sensors, this work demonstrates, through the case study of the SKU:SEN0193 sensor, the possibility of using low-cost technologies for monitoring environmental variables. Different combinations of sensors and microcontrollers, as well as their physical adaptation, can be used in order to improve their accuracy and durability. However, further research on this topic would enable the expansion of environmental monitoring in regions with budget limitations.

## Figures and Tables

**Figure 1 sensors-23-02451-f001:**
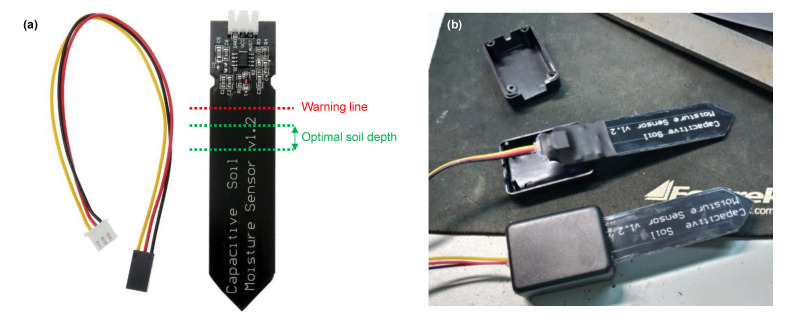
(**a**) The optimal soil contact zone for the SKU sensor for best response and (**b**) protection of its electronic components.

**Figure 2 sensors-23-02451-f002:**
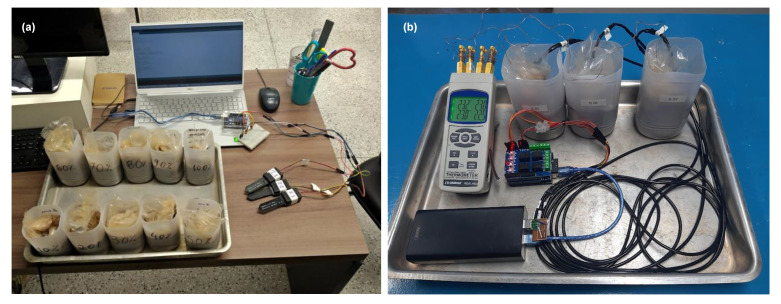
Lab conditions during (**a**) calibration and (**b**) temperature tests.

**Figure 3 sensors-23-02451-f003:**
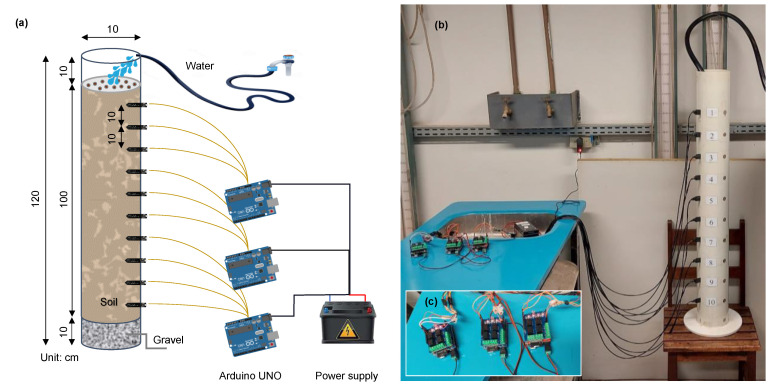
(**a**) Outline of column tests, (**b**) infiltration column, and (**c**) Arduino controllers and shields.

**Figure 4 sensors-23-02451-f004:**
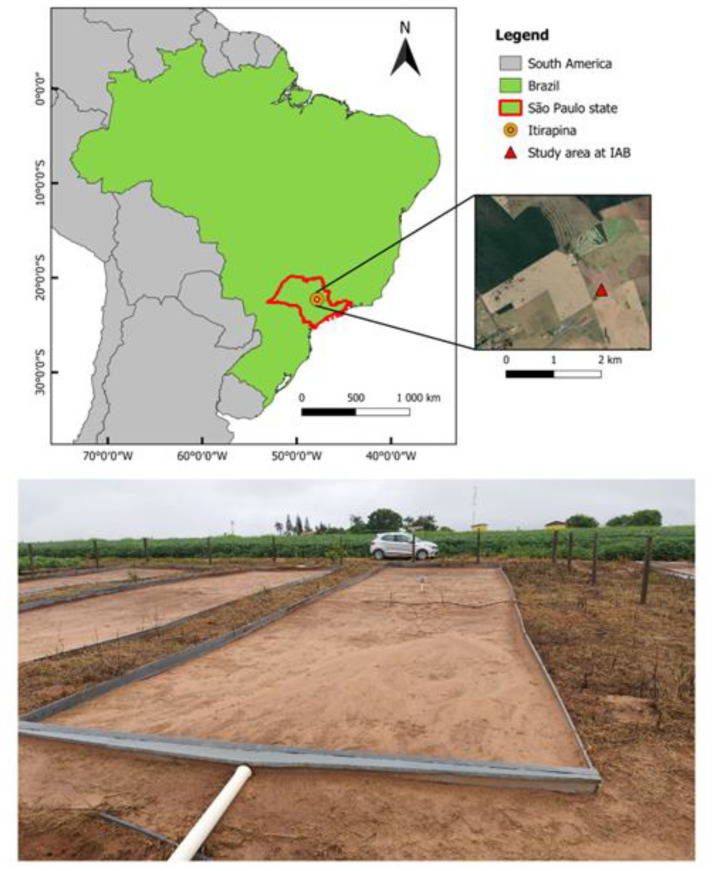
Location of the experimental field site.

**Figure 5 sensors-23-02451-f005:**
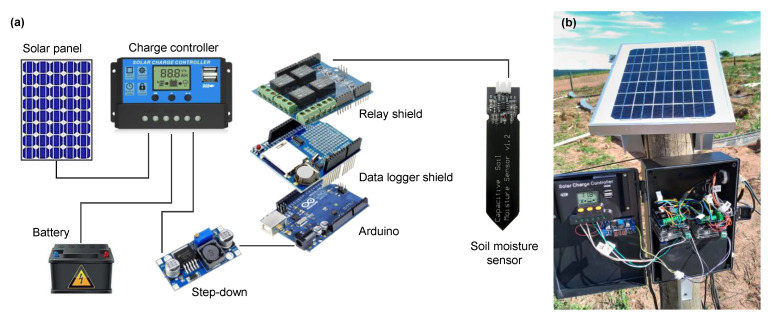
(**a**) Components scheme and (**b**) photo of the soil moisture monitoring station.

**Figure 6 sensors-23-02451-f006:**
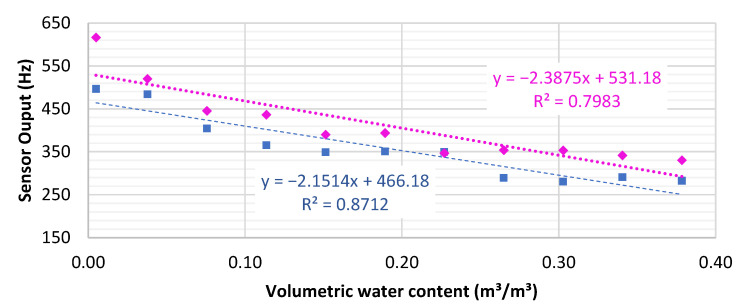
Sensor response and linear calibration curves under 3.3 V (pink dots and curve) and 5.5 V (blue dots and curve) input voltage.

**Figure 7 sensors-23-02451-f007:**
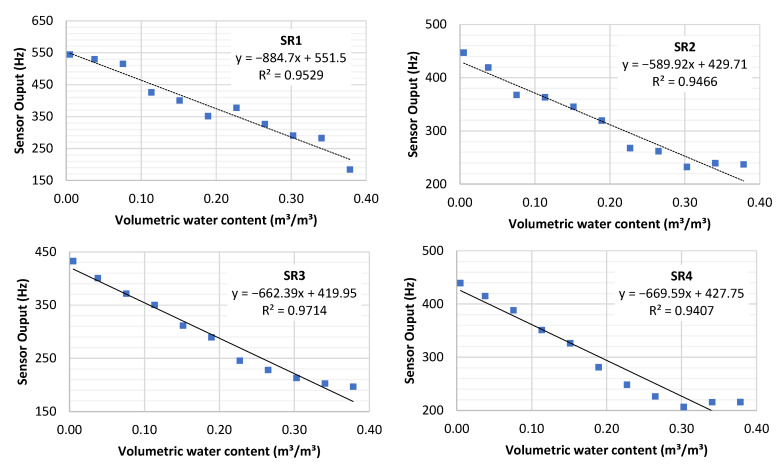
Individual sensor calibration curves used in field testing.

**Figure 8 sensors-23-02451-f008:**
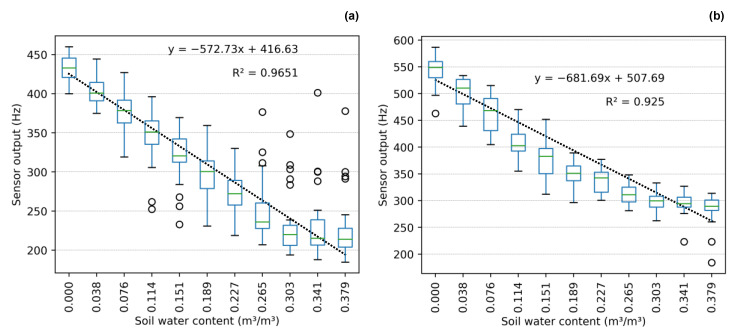
Single-point calibration curves for (**a**) group 1 and (**b**) group 2.

**Figure 9 sensors-23-02451-f009:**
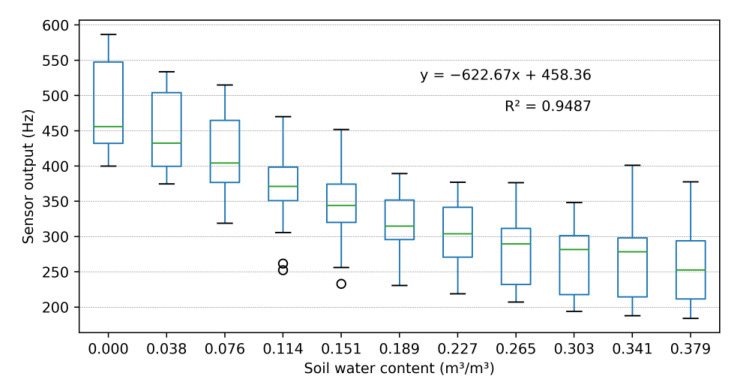
Universal sensor calibration curve.

**Figure 10 sensors-23-02451-f010:**
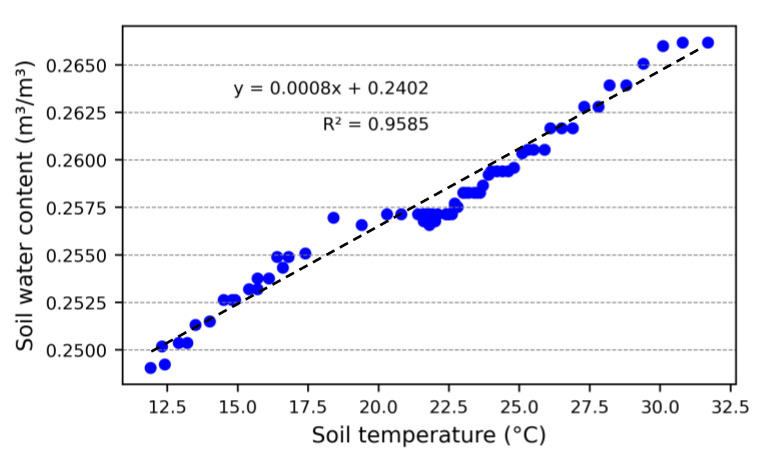
Sensor response under temperature variation.

**Figure 11 sensors-23-02451-f011:**
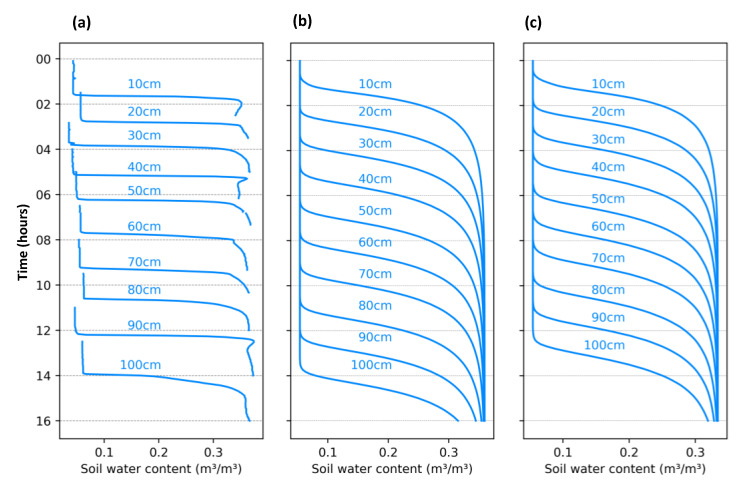
(**a**) Soil water content measured by sensors and simulated through the (**b**) Hydrus model before and (**c**) after its calibration for the soil column.

**Figure 12 sensors-23-02451-f012:**
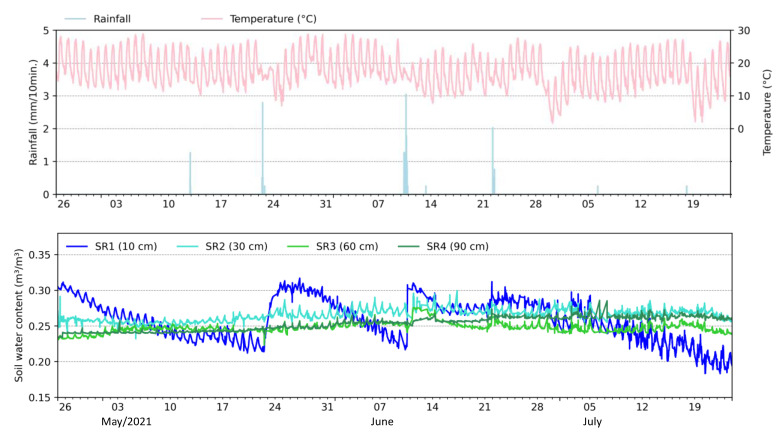
Observed soil moisture oscillations under field conditions.

**Table 1 sensors-23-02451-t001:** Soil properties at 14, 30, 60, and 90 cm depth in the study area.

Depth (cm)	Texture (Weight %)			BD (g cm^−3^)	PD (g cm^−3^)	OM (g dm^−3^)	CEC
	Clay	Silt	Sand				
0−14	12	3	85	1.43	2.64	23	36
30	12	6	81	1.49	2.64	10	24
60	10	5	85	1.59	2.65	19	28
90	15	1	84	1.52	2.65	8	20

where BD is bulk density, PD is particle density, OM is organic matter content, and CEC is cation exchange capacity.

**Table 2 sensors-23-02451-t002:** Input parameters of Hydrus-1D model.

Parameters	Values/Condition
*Geometry information*	
Depth (cm)	100
Mesh size (cm)	1
Number of layers	1
*Time information*	
Simulation time (h)	16
Time step	1 h
*Hydraulics properties*	
Sand (%)	81
Silt (%)	6
Clay (%)	12
Bulk density (g cm^−3^)	1.43
$\theta_{r}$ (cm^3^ cm^−3^)	0.0524
$\theta_{r}$ (cm^3^ cm^−3^)	0.376
α **(cm^−1^)**	0.0362
n (-)	1.438
$K_{s}$ (cm d^−1^)	10.768
** L **	0.5
*Boundary conditions*	
Upper boundary condition	Atmospheric BC with surface layer
Lower boundary condition	Free drainage
*Variable boundary conditions*	9.88 mm/h

**Table 3 sensors-23-02451-t003:** Material used for building the wetting front monitoring station.

Material	Quantity	Cost per Unit
Capacitive soil moisture sensor SKU:SEN0193 v1.2	4 units	BRL 28.90/USD 5.4
Jumpers (male and female)	20 units	BRL 2.79/USD 0.52
Arduino Uno R3	1 unit	BRL 89.90/USD 16.80
Relay shield 5V 4 channels	1 unit	BRL 42.65/USD 7.97
Datalogger shield	1 unit	BRL 59.90/USD 11.20
Memory card 8 gb	1 unit	BRL 39.50/USD 7.38
Step down LM2596S	1 unit	BRL29.99/USD 5.61
Battery 12v 7a	1 unit	BRL 69.90/USD 13.07
Solar panel 60 W	1 unit	BRL 275.00/USD 51.40
Charge controller 30a	1 unit	BRL 62.00/USD 11.59
Electrical box 170 × 120 × 90 mm	4 unit	BRL 45.34/USD 8.47
Electrical box 22 × 33 × 46 mm	1 unit	BRL 4.30/USD 0.80
Heat shrink tubing 18.00 mm^2^	15 cm	BRL 12.90/USD 2.41
Silicone transparent	1 tube	BRL 19.90/USD 3.72
		Total: BRL 869.67/USD 162.56

**Table 4 sensors-23-02451-t004:** Statistical metrics for different calibration methods.

	Individual	Single Point	Universal
**R^2^**	0.87–0.97	0.92 (group 1)–0.96 (group 2)	0.95
**RMSE (cm^3^.cm^−3^)**	0.054–0.078	0.061 (group 1)–0.092 (group 2)	0.082

**Table 5 sensors-23-02451-t005:** Qualitative comparison of different soil moisture sensors.

Sensor: Manufacturer	Cost	Accuracy	QualifiedLabor Demand	Sampling Volume	LifeExpectancy
CS650: Campbell Scientific	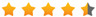		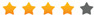	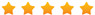	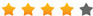
ECH2O 10HS: Decagon	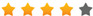	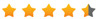		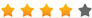	
ECH2O EC-5: Decagon	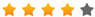	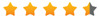	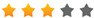	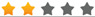	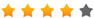
SM150T/ML3: Delta-T Devices	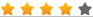	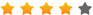	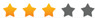		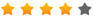
SoilVUE10: Campbell Scientific	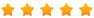	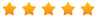	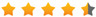		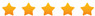
TEROS-10: Edaphic scientific	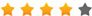	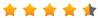	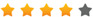	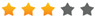	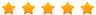
TDR-315H: Acclima company	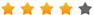		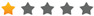	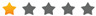	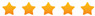
TRIME-PICO 64: Imko	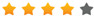	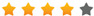	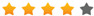	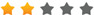	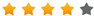
SKU:SEN0193: DFRobot	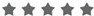	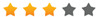	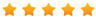	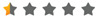	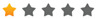

**Table 6 sensors-23-02451-t006:** Accuracy of different soil moisture sensors.

Sensor: Manufacturer	Accuracy (RMSE)	Reference
CS655: Campbell Scientific	±0.017 m^3^ m^−3^	[51]
ECH2O 10HS: Decagon	±0.031 m^3^ m^−3^	[52]
ECH2O EC-5: Decagon	±0.028 m^3^ m^−3^	[53]
ECH2O EC-5: Decagon	±0.017 m^3^ m^−3^	[51]
ECH2O 5TE: Decagon	±0.026 m^3^ m^−3^	[54]
ECH2O 5TE: Decagon	±0.05 m^3^ m^−3^	[55]
SoilVUE10: Campbell Scientific	±0.01 m^3^ m^−3^	[56]
SM150T/ML3: Delta-T Devices	±0.03 m^3^ m^−3^	[57]
TDR-315H: Acclima company	±0.013 m^3^ m^−3^	[51]
TEROS-12: Edaphic scientific	±0.015 m^3^ m^−3^	[51]
TRIME-PICO 64: Imko	±0.03 m^3^ m^−3^	[58]
SKU:SEN0193: DFRobot	±0.067 m^3^ m^−3^	[24]
SKU:SEN0193: DFRobot	±0.08 m^3^ m^−3^	[42]

## Data Availability

The dataset used in this research is available upon valid request to any of the authors of this research article.

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
