# Peer review of "Automated Low-Cost Soil Moisture Sensors: Trade-Off between Cost and Accuracy"

_sensors, 2023, doi:10.3390/s23052451_

Round 1

Reviewer 1 Report

The aim of the paper “Automated Low-Cost Soil Moisture Sensors: Trade-Off Between Cost and Accuracy” could be interesting for the readers of Sensors.

The objective of the study is clear. The use this low-cost soil moisture sensors could be indicated for limited-budget projects.  The calibrate methods used are appropriate and data support conclusions.

Generally speaking, none comparison has been done with other sensors, that they are called by authors commercial sensor with higher cost and higher accuracy. This paper only shows a relative comparison in a graphic way, where any figure about accuracy for commercial sensor have been shown. So, there is a lack of information in order to be able to do a comparison between low cost sensors and commercial sensor related to the accuracy.

Secondly, soil moisture has not clearly defined. There is a lack of information about several meanings, such as soil saturation, wet soil, dry soil. Soil moisture in soil science can be expressed in gravimetric or volumetric water content in percentages, or even thorough its soil water potentials in kPa. So, when a sensor needs to be calibrated I suppose it was going to be compared with objective water contents.

Thirdly, as a consequence of the above, field capacity water content or wilting coefficient water content are unknown in terms of the sensor outputs.

Fourthly, a big quantity of references inside the text are lost. I would recommend that authors use the same way for all bibliographic citations.

And finally, a homogeneous sandy soil has been used in this calibration. I wonder how  will the functioning be in other soils and why this point has not been talking about neither introduction nor concussions.

Author Response

Reviewer #1

Question: The aim of the paper “Automated Low-Cost Soil Moisture Sensors: Trade-Off Between Cost and Accuracy” could be interesting for the readers of Sensors. The objective of the study is clear. The use this low-cost soil moisture sensors could be indicated for limited-budget projects.  The calibrate methods used are appropriate and data support conclusions.

Answer: Thank you very much for your helpful comments and suggestions which significantly improved the quality of our paper. Please see below the answer (in green colour) to each comment made. We hope that this version reaches the “Sensors” high standard for publication. Note that the lines mentioned refers to the clean version of the paper.

Question: Generally speaking, none comparison has been done with other sensors, that they are called by authors commercial sensor with higher cost and higher accuracy. This paper only shows a relative comparison in a graphic way, where any figure about accuracy for commercial sensor have been shown. So, there is a lack of information in order to be able to do a comparison between low cost sensors and commercial sensor related to the accuracy.

We are grateful for your concern about the proper comparison with commercial sensors. Of course, the study would have been better if we could compare different techniques side by side under identical conditions. However, due to financial limitations this was not possible. Even though we do not provide direct comparison with commercial sensors, we included a new Table 6, line 441) that summarizes previous studies addressing the accuracy of soil moisture sensor. Please also note the discussion about this matter between lines 433-455.

Question: Secondly, soil moisture has not clearly defined. There is a lack of information about several meanings, such as soil saturation, wet soil, dry soil. Soil moisture in soil science can be expressed in gravimetric or volumetric water content in percentages, or even thorough its soil water potentials in kPa. So, when a sensor needs to be calibrated I suppose it was going to be compared with objective water contents.    

Answer: Thank you very much for your helpful observation. Our first results were given in terms of soil saturation (percentage of the voids, spaces, and cracks filled with water) since it is a common unit used in agriculture. However, we agreed that our results would be more fruitful to be presented in terms of gravimetric water content (volume of water per dry soil), then, the entire paper was reformulated to this unit (m³/m³). Please see bellow our definitions for the terms used:

Soil water content (SWC): volume of water divided by the dry bulk soil volume (see lines 31-32);

Dry soil condition: Soil samples dried at 105ºC for 48 h (lines 125-126);

Saturated soil condition: Soil samples where all the pores (empty spaces between the solid soil particles) are filled with water (128-129).

Question: Thirdly, as a consequence of the above, field capacity water content or wilting coefficient water content are unknown in terms of the sensor outputs.

Answer: This was another reason that we believed that it would be interesting to have data presentation in terms of volumetric soil water content (m³/m³).

Question: Fourthly, a big quantity of references inside the text are lost. I would recommend that authors use the same way for all bibliographic citations.

Answer: Some editing problem must have happened when including the text in the template, but we double-checked the references, and everything looks fine in the new version of the manuscript. Thank you for noticing.

Question: And finally, a homogeneous sandy soil has been used in this calibration. I wonder how will the functioning be in other soils and why this point has not been talking about neither introduction nor concussions.

Answer: As you mentioned, the authors agree that soil heterogeneity affects the spatial behavior of the vertical (re)distribution of water in the vadose zone (see introduction session, lines 35-40). The use of a homogenous sandy soil is a limitation to our findings (see discussion session, lines 463-467) but in the present study we did not test different soil types since it would interfere with the research scope to evaluate the sensor variability output under same conditions for later development of universal calibration curve. Our findings are specific to the pedological characteristics of the soil sample tested. Previous studies [1–4] also indicated the importance of constructing local calibration curves since the sensor is sensitive to soil texture and bulk density. [4] for example, tested the SKU sensor under three soil textures and found distinct calibration curves (Lines 363-367).  We are not sure about how significant the impact of the soil texture has on sensor output when applying the universal or single-point calibration curve and future studies are then required (see lines 374-376).

References

  1. Kulmány, I.M.; Bede-Fazekas, Á.; Beslin, A.; Giczi, Z.; Milics, G.; Kovács, B.; Kovács, M.; Ambrus, B.; Bede, L.; Vona, V. Calibration of an Arduino-Based Low-Cost Capacitive Soil Moisture Sensor for Smart Agriculture. J. Hydrol. Hydromechanics 2022, 70, 330–340, doi:10.2478/johh-2022-0014.
  2. Souza, G.; De Faria, B.T.; Gomes Alves, R.; Lima, F.; Aquino, P.T.; Soininen, J.P. Calibration Equation and Field Test of a Capacitive Soil Moisture Sensor. In Proceedings of the 2020 IEEE International Workshop on Metrology for Agriculture and Forestry, MetroAgriFor 2020 - Proceedings; Institute of Electrical and Electronics Engineers Inc., November 4 2020; pp. 180–184.
  3. López, E.; Vionnet, C.; Ferrer-Cid, P.; Barcelo-Ordinas, J.M.; Garcia-Vidal, J.; Contini, G.; Prodolliet, J.; Maiztegui, J. A Low-Power IoT Device for Measuring Water Table Levels and Soil Moisture to Ease Increased Crop Yields. Sensors (Basel). 2022, 22, 1–23, doi:10.3390/s22186840.
  4. Pereira, R.M.; Sandri, D.; Silva Júnior, J.J. da Evaluation of Low-Cost Capacitive Moisture Sensors in Three Types of Soils in the Cerrado, Brazil. Rev. Eng. na Agric. - REVENG 2022, 30, 262–272, doi:10.13083/reveng.v30i1.14017.
  5. Placidi, P.; Gasperini, L.; Grassi, A.; Cecconi, M.; Scorzoni, A. Characterization of Low-Cost Capacitive Soil Moisture Sensors for IoT Networks. Sensors 2020, 20, 3585, doi:10.3390/s20123585.

Reviewer 2 Report

The work is interesting because the authors propose the use of low-cost technology to monitor moisture in the soil profile. I recommend that the authors continue with the work on more complex soils with greater pedogenetic development.

Author Response

Reviewer #2

Question: The work is interesting because the authors propose the use of low-cost technology to monitor moisture in the soil profile. I recommend that the authors continue with the work on more complex soils with greater pedogenetic development.

Answer: We are very grateful for reviewing and supporting our publication. Please see below the answer (in green colour) to each comment made. Note that the lines mentioned refers to the clean version of the paper.

As you mentioned, the authors agree that soil heterogeneity affects the spatial behavior of the vertical (re)distribution of water in the vadose zone (introduction session, lines 35-37). The use of homogenous sandy soil is a limitation to our findings (discussion session, lines 464-466) but we could not be testing different soil types once it would interfere in the research scope to evaluate the sensor variability output under same conditions for later development of universal calibration curve. Our findings are specific to the pedological characteristics of the soil sample tested. Previous studies [1–4] also indicated the importance of constructing local calibration curves since the sensor is sensitive to soil texture and bulk density. [4] for example, tested the SKU sensor under three soil textures and found distinct calibration curves (Lines 363-367). We are not sure about how significant the impact of the soil texture has on sensor output when applying the universal or single-point calibration curve and future studies are then required (see lines 374-376).

Question: Line 242. The use of this type of sensor lies in its difficulty in recording data above 82%, it rarely records values above 82%, even when it is inside a container with water. It must be interpreted that in values close to 82%, the soil of the horizon of the profile is already water saturation.

Answer: We believe that the reviewer misunderstands the concept of soil water saturation when stating “…even when it is inside a container with water”. Our first results were given in terms of soil saturation (percentage of the voids, spaces, and cracks filled with water) since it is a common unit used in agriculture. However, we believe that our results would be more understandable if presented in terms of gravimetric water content (volume of water per volume of dry soil), then, the entire paper was reformulated to this unit (m³/m³). To clarify and makes readers more comfortable with soil science, we stated different concepts all over the text:

Soil water content (SWC): volume of water divided by the dry bulk soil volume (see lines 31-32);

Dry soil condition: Soil samples dried at 105ºC for 48 h (lines 125-126);

Saturated soil condition: Soil samples where all the pores (empty spaces between the solid soil particles) are filled with water (128-129).

Question: Line 292. Due to the clay content at different depths, it is not decisive that soil compaction has a direct influence. It is more probable that the disintegration of the particles in the superficial horizon is more significant than in subsurface horizons with a natural aggregation. The authors should mention how they determined the infiltration rate in the different genetic horizons, as well as their depth within the soil profile.

Answer: Thank you for you comment. Note that in the referred line 292 we are addressing the results found the repacked soil column under laboratory conditions, meaning that there is homogenous soil characteristics (lines 155-156). We acknowledge that under field conditions, the clay content at different depths plays a direct influence, but it not the case under those controlled situation mentioned before. Regarding the infiltration at a given depth, it is based in sensors’ output oscillation indicating the arrival of the wetting front. Following this, the infiltration rate is basically the ratio between the distance of sensors (10 cm) and time taken to flow between the area where those sensors are installed (lines 302-303).

 Question: Line 405. It is recommended that the authors mention the instruments used to measure the degree of compaction of the various horizons to indicate that the degree of compaction was decisive at the time of calibration and operation at the field level.

Answer: We did not use a specific instrument to measure the degree of compaction during calibration, we had a known mass of dry soil that was compacted to a pre-defined volume to reach the desired bulk density (see lines 125-127). Previous studies have already indicated that the degree of soil compaction used during calibration induces a significant difference in the frequency response of the sensor (as also found by [5] and [2]) and since field soil is not vertically homogeneous having different bulk densities (Table 1, line 119) it will interfere in the sensor output.

References

  1. Kulmány, I.M.; Bede-Fazekas, Á.; Beslin, A.; Giczi, Z.; Milics, G.; Kovács, B.; Kovács, M.; Ambrus, B.; Bede, L.; Vona, V. Calibration of an Arduino-Based Low-Cost Capacitive Soil Moisture Sensor for Smart Agriculture. J. Hydrol. Hydromechanics 2022, 70, 330–340, doi:10.2478/johh-2022-0014.
  2. Souza, G.; De Faria, B.T.; Gomes Alves, R.; Lima, F.; Aquino, P.T.; Soininen, J.P. Calibration Equation and Field Test of a Capacitive Soil Moisture Sensor. In Proceedings of the 2020 IEEE International Workshop on Metrology for Agriculture and Forestry, MetroAgriFor 2020 - Proceedings; Institute of Electrical and Electronics Engineers Inc., November 4 2020; pp. 180–184.
  3. López, E.; Vionnet, C.; Ferrer-Cid, P.; Barcelo-Ordinas, J.M.; Garcia-Vidal, J.; Contini, G.; Prodolliet, J.; Maiztegui, J. A Low-Power IoT Device for Measuring Water Table Levels and Soil Moisture to Ease Increased Crop Yields. Sensors (Basel). 2022, 22, 1–23, doi:10.3390/s22186840.
  4. Pereira, R.M.; Sandri, D.; Silva Júnior, J.J. da Evaluation of Low-Cost Capacitive Moisture Sensors in Three Types of Soils in the Cerrado, Brazil. Rev. Eng. na Agric. - REVENG 2022, 30, 262–272, doi:10.13083/reveng.v30i1.14017.
  5. Placidi, P.; Gasperini, L.; Grassi, A.; Cecconi, M.; Scorzoni, A. Characterization of Low-Cost Capacitive Soil Moisture Sensors for IoT Networks. Sensors 2020, 20, 3585, doi:10.3390/s20123585.

Round 2

Reviewer 1 Report

I have no any more clarifications to do.